# Pulmonary Metastasectomy: A Multicenter Comparison of Wedge Resection Versus Anatomic Resection for Single Metastases of Colorectal Cancer

**DOI:** 10.3390/cancers17203299

**Published:** 2025-10-11

**Authors:** Stefan Welter, Isabelle Moneke, Ramzi Wara, Antonia Uyen-Thao Le, Ahmad Shalabi, Thomas Graeter, Till Ploenes, Daniel Baum

**Affiliations:** 1Department of Thoracic Surgery, Lungenklinik Hemer, 58675 Hemer, Germany; ramzi.wara@outlook.de; 2Department of Thoracic Surgery, Medical Center-University of Freiburg, Faculty of Medicine, University of Freiburg, 79106 Freiburg im Breisgau, Germany; isabelle.moneke@uniklinik-freiburg.de (I.M.); antonia.uyen-thao.le@uniklinik-freiburg.de (A.U.-T.L.); 3Department of Thoracic Surgery, SLK Lungenklinik Löwenstein, 74245 Löwenstein, Germany; ahmad.shalabi@slk-kliniken.de (A.S.); thomas.graeter@slk-kliniken.de (T.G.); 4Department of Thoracic Surgery, Fachkrankenhaus Coswig, Lung Center, 01640 Coswig, Germany; till.ploenes@ukdd.de (T.P.); daniel.baum@lungenzentrum-coswig.de (D.B.); 5Division of Thoracic Surgery, Department of Visceral, Thoracic and Vascular Surgery, Faculty of Medicine and University Hospital Carl Gustav Carus, 01069 Dresden, Germany

**Keywords:** pulmonary metastases, local margin recurrence, anatomical lung resection, lymph node recurrence, safety margins

## Abstract

Operative treatment of single pulmonary metastasis from colorectal cancer should be focused on the prevention of local margin recurrence. With a retrospective evaluation from 4 German thoracic surgery centers, we could demonstrate that anatomical metastasis resection is associated with larger safety margins, greater number of lymph nodes removed, no incomplete resections and fewer intrapulmonary margin recurrences, compared to non-anatomic resections like wedge or laser enucleation. Overall (OS) and recurrence-free survival (RFS) were excellent, but were not different between the groups. After propensity-score matching in patients with peripheral or intermediate lesions, anatomic resection was associated with a significantly lower rate of local margin recurrence, no significant difference in intrathoracic lymph node recurrence and no difference in OS or RFS was found.

## 1. Introduction

Pulmonary metastasectomy has become an important intervention for patients with stage IV extrapulmonary solid tumors in carefully selected cases [1], offering the potential to improve survival. A recent publication presented a significant survival advantage of surgical resection over radiation therapy and conservative management of solitary lung metastases from colorectal cancer (HR: 0.68; *p* < 0.001) [2]. However, tumor recurrence after metastasectomy remains a common challenge. Well-established predictors of improved post-metastasectomy survival include metachronous tumor evolution, the presence of only a single metastasis and complete resection [3]. Based on these factors, it is possible to identify a subgroup of patients—those with metachronous single metastasis and a long disease-free interval—who may have a genuine chance for cure. These patients might benefit from a more radical local treatment aimed at preventing local recurrence after metastasectomy.

Anatomical resection (such as lobectomy or segmentectomy) instead of wedge resection may offer three key advantages for these patients:*Greater safety margins:* Anatomical resection provides wider margins, which should reduce the risk of local intrapulmonary recurrence at the resection margin [4].*Lymph node clearance:* Removal of intralobar or intrasegmental lymphoid tissue may include undetected lymph node metastases or lymphatic involvement, thereby reducing rates of locoregional recurrence.*Reduced occult metastases:* The larger volume of tissue removed decreases the likelihood of occult metastases in healthy lung tissue, potentially lowering the risk of new intrapulmonary metastases.

These hypotheses are supported by several retrospective studies. Non-anatomical resection of single colorectal carcinoma (CRC) lung metastases has been associated with a higher rate of local margin recurrence [5], and intrapulmonary recurrence has been found to be reduced following segmentectomy rather than wedge resection [6]. Furthermore, patients undergoing major anatomical resection for CRC lung metastases have been found to have significantly longer disease-specific survival (DSS) and disease-free survival (DFS) in another large multicentre study [7].

Additional studies, such as those by Prisciandaro et al. [8], have explored the impact of resection extent on postoperative outcomes, concluding that anatomical resections are associated with improved recurrence-free survival (RFS).

In summary, current evidence suggests that anatomical resection may help keep local margin recurrence rates low. However, as highlighted in an editorial [4], the retrospective studies on pulmonary metastasectomy often suffer from non-systematic and incomplete reporting, which limits the strength of these recommendations. There is a clear need for more systematic and standardized reporting to enable better comparison across studies.

### Study Rationale and Objectives

With this in mind, our study aims to provide a comprehensive multicenter comparison between two primary surgical techniques: non-anatomic resection and anatomic resection for single CRC lung metastases.

## 2. Materials and Methods

The study was approved by the Ethics Commission of the Medical Association Westphalia-Lippe (2024-084-f-S, 14 September 2024). Due to the retrospective nature of the study, written informed consent from patients was waived. The surgery was performed in accordance with local standards and the assessment of the responsible surgeon. As only one lesion had to be removed, sufficient safety margins were prioritised over tissue preservation. When the laser was used, the surgeon stated that complete resection had been achieved. If R1 was stated by the pathologist, it was included in the database even after laser resection. An anatomic resection was performed when the lesion was larger than 20 mm or centrally located.

All participating thoracic surgery departments have their own prospective metastases database, including every surgical intervention for pulmonary metastases. A subset of all patients with single pulmonary metastases from CRC who underwent resection was selected from the local databases. Follow-up with a focus on local intrapulmonary recurrence was retrieved from medical records and local tumor documentation systems. Each center received a standardised entry form containing all the relevant parameters and submitted every case involving resection of single colorectal cancer (CRC) metastases after anonymisation. All anonymised local data were then consolidated into one entry mask containing the following parameters:

Age and gender; date of primary treatment; primary tumor stage; disease-free interval (DFI); date, access and mode of metastasis surgery; position and size of the metastasis; safety distance; mode and number of dissected lymph nodes; and number of resected nodes. Complications were classified according to the Clavien–Dindo classification and follow-up information included date of death or last information alive and the status of tumor. The size of the metastasis was measured by its maximum diameter, and the safety margin was defined as the minimum amount of healthy lung tissue surrounding the metastasis nodule, as determined by the pathologist. Anatomical resections included segmentectomies, lobectomies and bilobectomies. Non-anatomical resections comprised wedge resection, laser, or cautery enucleation. DFI was calculated as the time from definitive primary tumor treatment to pulmonary metastasectomy. Local recurrence was defined as growing solid tissue at the site of the former resection margin, presence of scar formation between the nodule and visceral pleura as residual from a prior surgery, the nodule located in the same segment of a former metastasectomy or metal remnants in the lesion after a former wedge resection with staplers or intrapulmonary marking with clips [9]. Follow-up was usually performed in colorectal cancer centers, but patients were asked to provide their chest CT scans to the thoracic surgery department. All follow-up CT scans indicating a recurrence of metastasis in the lung were reviewed by an experienced surgeon from each contributing site. Unfortunately, due to the retrospective nature of this analysis, it is not possible to re-examine every patient systematically. Written CT reports from radiologists stating “no tumor recurrence” in the chest were accepted.

### Statistical Evaluation

All analyses were performed using R software (version 4.5.1; R Foundation for Statistical Computing, Vienna, Austria), utilizing specialized packages including survival, survminer, MatchIt, rstatix, and ggplot2. As the study was retrospective, no sample size calculation was performed in advance. Descriptive statistics were applied to summarize patient demographics, tumor characteristics, and perioperative variables. Continuous variables were expressed as means with standard deviations (SD) or medians with interquartile ranges (IQR), depending on distribution, which was assessed using the Jarque–Bera test for normality. Categorical variables were reported as absolute and relative frequencies. Survival analysis was performed using the Kaplan–Meier method, with overall survival (OS) and recurrence-free survival (RFS) estimated from the time of metastasectomy to death, recurrence, or last follow-up. Differences in survival between groups were compared using the log-rank test.

To minimize potential selection bias and confounding inherent in retrospective multicenter data, propensity score matching was employed. Matching was performed using the nearest-neighbor method without replacement at a 1:1 ratio, balancing covariates such as age, sex, tumor size, disease-free interval (DFI), lymph node involvement, surgical approach (open vs. minimally invasive), and safety margins. Balance diagnostics were assessed using standardized mean differences, and matched groups were compared to evaluate the impact of resection type (anatomic vs. non-anatomic) on outcomes.

For comparisons of categorical variables between groups, Fisher’s exact test was used due to small sample sizes in some subgroups. Continuous variables were analyzed using either Student’s *t*-test or the Mann–Whitney U test, as appropriate. Hazard ratios (HRs) with corresponding 95% confidence intervals (CIs) were calculated where applicable. Statistical significance was defined as a two-sided *p* value < 0.05. All visualizations, including Kaplan–Meier survival curves and covariate balance plots, were generated using ggplot2, survminer, and related R packages to illustrate key findings.

Propensity score matching was performed for 112 cases with intermediate and peripheral metastases using the significant differences (*p* < 0.1) between the anatomic and non-anatomic groups as matching parameters.

GenAI (ChatGPT online and Deepl write) was used for language and grammar improvement.

## 3. Results

A total of 181 cases were collected from four different thoracic surgery centres (Hemer *n* = 67, Coswig *n* = 57, Freiburg *n* = 36, Löwenstein *n* = 21). Following the removal of fifteen cases due to an absence of follow-up data, the analysis was further refined to encompass 166 patients. The demographic data are illustrated in Table 1.

The primary tumor was located more frequently in the rectum (59%) than in the colon (41%), and the cohort comprised more men (62.6%) than women (37.4%). The median age of the subjects was 69 years (SD ± 10.1; range 39.1–92.6). The majority of lesions manifested as metachronous (86.7%), and a smaller cohort of patients had previously undergone resection for liver metastases (22.3%). The preferred approach was open thoracotomy (75.9%), with lobectomy (25.3%), enucleation (24.7%) and segmentectomy (21.7%) being used for metastasis removal in most cases. For the purpose of subsequent comparison, 93 anatomic resections (56%) and 73 non-anatomic resections (44%) were collated. Major postoperative complications (Clavien–Dindo IIIa and IIIb) occurred in 4.2% of patients. No instances of Grade IV were observed, and postoperative mortality remained at 0%.

*Pathology:* The maximum size of the metastases was median 21.0 mm (range 4–115) and the minimum safety margin was median 5.0 mm (range 0–50). The complete resection (R0) rate was 95.8%, with 4.2% of cases being designated as R1.

*Lymph node removal:* Lymph node removal was performed in 82.4% of the interventions, with a median number of 4 (range: 0–42) nodes removed. Lymph node sampling, radical dissection or no removal was performed in 79/165 (48%), 57/165 (35%) and 29/165 (18%) of cases, respectively. In a total of 16 (9.6%) cases, lymph node involvement was detected.

*Tumor recurrence:* At the time of data amalgamation, 64 out of 166 patients (38.6%) had died. A total of 87 out of 161 patients (54%) experienced tumor recurrence at any site, and 62 out of 161 patients (38.5%) experienced tumor recurrence within the lung. Intrapulmonary margin recurrence was detected in 25 out of 145 (17.2%) cases, while intrathoracic lymph node recurrence was found in 14 out of 138 (10.1%) cases. Percentages were calculated based on the number of patients with valid data for each parameter.

*Survival:* The 5- and 10-year overall survival (OS) rates were 69.9% and 47.3%, respectively, and the corresponding recurrence-free survival (RFS) rates were 59.3% and 46.3% (Figure 1). Local intrapulmonary margin recurrence appeared after a mean of 2.6 (median 1.6) years; only 4 of 24 (16%) appeared 5 years or later after metastasectomy. The 5- and 10-year survival rates without local margin recurrence were 88.5% and 62.1%, respectively.

Any tumor recurrence was associated with worse overall survival (OS) (median 5.8 vs. 15.2 years; *p* < 0.001). Intrapulmonary tumor recurrence was also associated with worse OS (median 6.3 vs. 15.2 years; *p* = 0.049). Finally, intrapulmonary margin recurrence was associated with a non-significant decrease in OS (median 7.8 vs. 11.4 years; *p* = 0.264) (Figure 2).

*Risk factors for local recurrence:* Intrapulmonary local margin recurrence was found to be more prevalent in non-anatomic (25.4%) compared to anatomic (11.6%) resections (*p* = 0.052) (Figure 3). Furthermore, this recurrence was observed to be more prevalent in cases where no lymph nodes were removed (24%) compared to lymph node sampling (22.4%) or radical lymph node dissection (7.6%) (*p* = 0.063). Furthermore, local margin recurrence was found to be associated with small safety margins (*p* < 0.001), a small number of lymph nodes removed (*p* < 0.001) and with intrathoracic lymph node recurrence (*p* = 0.001) (Table 2).

Fisher’s exact test (*p* = 0.052) comparing the frequency of intrapulmonary local margin recurrence after non-anatomical with anatomical resections.

*Risk factors for intrathoracic lymph node recurrence*: Risk factors for the occurrence of lymph node recurrence are depicted in Table 3. There was an obvious association with intrapulmonary tumor recurrence (*p* < 0.001) and also with intrapulmonary margin recurrence (*p* < 0.001). Of the 14 cases, only one was detected with isolated lymph node recurrence. The other 13 cases also had intrapulmonary or extrapulmonary recurrence.

*Anatomic* vs. *non-anatomic:* Anatomic resection was performed for central lesions in 39/52 (75%) of cases, and local margin recurrence was registered in 6/38 (15.8%) of cases. Non-anatomic resection was performed for central lesions in 13/52 (25%), and local margin recurrence was registered in 4/11 (36.4%) (Chi^2^, *p* = 0.136). Anatomic resection was performed for peripheral and intermediate lesions in 54/112 (48.2%) cases, and local margin recurrence was registered in 4/51 (7.8%) cases. Non-anatomic resection was performed in 58/112 (51.8%) cases, and local margin recurrence was documented in 11/53 (20.8%) cases. The difference was not significant (Chi^2^, *p* = 0.061).

Compared with non-anatomic resection, anatomic resection was associated with clear margins (R0 = 100%) (*p* = 0.007), wider safety margins (*p* < 0.001), more frequent radical lymph node dissection (*p* < 0.001), a greater number of lymph nodes removed (*p* < 0.001), a higher detection rate of lymph nodal involvement (N1, N2) (*p* = 0.027), more central metastases (*p* = 0.053), larger metastases (*p* = 0.006) and fewer intrapulmonary local recurrences (*p* = 0.052) (Table 4, Figure 3).

*Propensity score matching:* Given that a central location of metastases frequently necessitates anatomic resections, a reasonable comparison with wedge resection is not useful. The objective of this study was to calculate the real possible benefit of anatomic versus non-anatomic resection. To this end, the risk factors for local intrapulmonary margin recurrence were recalculated after the exclusion of all centrally located metastases. With regard to the remaining 112 intermediate and peripheral metastases, local intrapulmonary recurrence was found to be significantly more prevalent following non-anatomic resection in comparison to anatomic resection, upon the implementation of propensity score matching (PSM) (OR 2.63; *p* = 0.042). No significant difference could be found concerning intrathoracic lymph node recurrence (OR 1.68; *p* = 0.392) (Table 5).

## 4. Discussion

The primary objective of this study was to determine whether radical anatomic resection of single colorectal cancer (CRC) lung metastases offers superior outcomes compared to atypical resection, specifically in terms of local intrapulmonary margin recurrence, intrathoracic lymph node recurrence, recurrence-free survival (RFS), and overall survival (OS). To date, this question has remained unanswered in the literature [8]. A significant challenge in previous studies has been the inconsistent reporting on metastasectomy procedures, which complicates the attribution of observed benefits solely to the mode of resection. Instead, advantages may be confounded by factors such as more comprehensive lymphadenectomies, larger safety margins, and the inclusion of multiple metastases [4]. This variability underscores the need for more precise and standardized research to clarify the true impact of surgical technique.

To address these limitations, we adopted a more rigorous study design. Our inclusion criteria focused exclusively on patients with a single, histologically confirmed CRC lung metastasis, ensuring a homogeneous patient population. This approach aligns with prior research emphasizing the prognostic significance of solitary metastases [6,10,11]. Additionally, we conducted a multi-center evaluation, systematically recording numerous confounding variables such as tumor size, location, timing of metastasis appearance (synchronous vs. metachronous), DFI and lymph node status, to enhance the robustness of our findings. Our hypothesis was that a solitary, metachronous CRC lung metastasis offers the greatest potential for cure, and thus, should be managed with an emphasis on complete resection and systematic lymph node dissection to prevent local margin recurrence—approaches supported by prior research emphasizing the importance of thorough oncologic clearance [6,12].

Data were collected from four specialized thoracic surgery centers, each with extensive experience—performing over 300 metastasectomies for various primaries within the past 10 years. Interestingly, despite this high volume, the number of patients with a single CRC lung metastasis was relatively small: 181 cases, of which 166 were suitable for final analysis. For context, previous large-scale studies such as Shiono et al. [6] included 553 patients undergoing their first CRC lung metastasectomy across 46 Japanese centers over four years, analyzing both single and multiple metastases. Hernández et al. [7] reported on 522 cases from 32 Spanish centers over two years, while Renaud et al. [5] documented 574 consecutive resections across French, Canadian, and Italian centers over a decade. To our knowledge, only one publication has focused exclusively on single CRC lung metastases, making our dataset a valuable addition to the literature [11,13]. Because of the inclusion of cases with multiple metastases in other studies, our focused cohort provides a clearer understanding of the outcomes specific to solitary metastases.

*Operative procedure*: Operative procedures were performed according to local standards, with no standardized protocol across centers. This resulted in variability in techniques, including the use of laser and minimally invasive approaches. Nonetheless, perioperative mortality was zero percent, and major complications were rare (4.2%), underscoring the safety and feasibility of surgical intervention for single lesions. Furthermore, every suspicious lesion was removed. In some cases, therefore, more than one specimen reached the pathologist. Ultimately, one metastasis was proven in each patient, with no evidence of any other malignancy. As 36 metastases were larger than 4 cm, and many metastases had a central position, the relatively high number of anatomical resections in our series may be explained.

*Anatomic vs. non-anatomic resection*: Renaud et al. [5] reported that non-anatomical resection was associated with a significantly higher rate of local margin recurrence compared to anatomical resection in patients with single colorectal carcinoma (CRC) lung metastases (4.8% vs. 54.2%, *p* < 0.001). Notably, this difference was observed only in patients harboring KRAS mutations, but not in wild-type patients.

Similarly, Shiono et al. [6] found a reduced rate of local intrapulmonary recurrence when CRC lung metastases were removed via segmentectomy instead of wedge resection (2.0% vs. 7.3%; *p* = 0.035). This reduction translated into better 5-year recurrence-free survival (RFS) rates—48.8% with segmentectomy versus 36.0% with wedge resection—and improved 5-year overall survival (80.1% vs. 68.5%).

Furthermore, Hernandez et al. [7], in a large multicenter Spanish study involving 522 patients with CRC lung metastases, compared 19.9% major anatomic resections (lobectomy, bilobectomy, pneumonectomy) with 80.1% lesser resections, including segmentectomies. They found that major resections were associated with significantly longer disease-specific survival (DSS) and disease-free survival (DFS), with hazard ratios (HR) of 0.6 and 0.5, respectively.

*Confounding factors:* Our primary aim was to minimize confounding factors as much as possible, which is why we focused our analysis specifically on CRC lung metastases. Notably, few studies have reported differences in disease-free survival (DFS) between primary tumor sites, indicating inferior DFS after pulmonary metastasectomy for rectal cancer compared to colon cancer [14,15]. Importantly, regarding local intrapulmonary margin recurrence, our data did not reveal any significant difference between the two primary sites, suggesting that surgical approach and margin status may be equally effective regardless of the primary tumor location.

*Oncologic outcome:* In terms of survival outcomes, our study demonstrates excellent 5- and 10-year overall survival (OS) rates of 70% and 47%, respectively, alongside impressive recurrence-free survival (RFS) rates of 59% at 5 years and 46% at 10 years. This highlights the potential for cure in this patient population. These results compare favorably with, or even surpass, those reported in previous studies [10,11,16], reinforcing the role of surgery as a highly effective treatment modality for carefully selected patients with solitary CRC lung metastases.

The survival free from local margin recurrence was similarly favorable, with 88.5% at 5 years and 62.1% at 10 years. It is well established that tumor recurrence significantly impacts survival; in our series, any recurrence (54%) was associated with a markedly reduced median survival (5.9 years versus 15.2 years; *p* < 0.001), aligning with findings from Forster et al. [17], Shiono et al. [18] and Suzuki et al. [15] who reported recurrence rates of approximately 57%, 56% and 69%, respectively. Others demonstrated that extrathoracic recurrence was significantly associated with worse prognosis than intrathoracic recurrence [19]. Risk factors for tumor recurrence are described: synchronous lung metastasis, previous resection of extrathoracic metastasis, increased uptake at FDG-PET scan and short (<12 months) DFI prior to lung metastasectomy [11,19] and elevated CEA [15].

*Local margin recurrence:* Specifically, intrapulmonary metastasis recurrence occurred in 38.5% of our patients, comparable to the 42.9% reported by Blackmon et al. [20]. Local margin recurrence was observed in 17.2% of cases, which aligns with previous reports—Shalabi et al. [21] documented a 10.2% rate following laser metastasectomy, while Shiono et al. [18] and Hernandez et al. [7] reported rates of 28% and 16.9% after wedge and anatomical resections, respectively. These findings suggest that achieving clear margins remains a critical factor, yet the variability indicates that other factors, such as tumor biology and systemic disease control, also influence recurrence risk. Interestingly, although local margin recurrence was associated with worse survival, this difference did not reach statistical significance in our cohort. This may be explained by the observation that some patients with local margin recurrences had repeat metastasectomy followed by long-term survival, and it may be explained by the fact that some of them experienced further distant metastases, which are known to have a more profound impact on prognosis and thus nullify the beneficial effect of anatomic resection. Consequently, the overall survival appears more heavily influenced by distant recurrence rather than local margin status alone, emphasizing the importance of comprehensive systemic management alongside surgical resection.

*Molecular alterations*: Molecular alterations in colorectal cancer (CRC) patients with pulmonary metastases may significantly influence surgical planning and outcomes. Notably, patients harboring mKRAS mutations have been shown to experience a higher rate of local recurrences when treated with wedge resection compared to anatomical resection (*p* = 0.001). This finding underscores the potential importance of molecular profiling in guiding surgical strategies, as the same difference was not observed in patients with wild-type KRAS [22]. Unfortunately, due to limited availability of molecular data in our cohort, we were unable to perform a comprehensive analysis of these genetic factors and their impact on recurrence patterns.

*Resection margins*: As previously emphasized in the literature [23,24,25], ensuring adequate surgical margins is crucial for minimizing local recurrence. In our study, local margin recurrence was also significantly associated with small safety margins (*p* < 0.001). The choice of resection method was primarily dictated by intraoperative assessment and surgeon preference, which resulted in a moderate local recurrence rate of 17.2% in our study. In our series, the number of anatomical resections was higher than that of non-anatomical resections (93 versus 73). Wedge resection is widely accepted as the standard method for pulmonary metastasectomy, mainly because the amount of resection is not related to survival in most cases. Wedge resection is also associated with lower morbidity, and lung-sparing resection allows for eventual redo surgery. Conversely, segmentectomy and even lobectomy are associated with lower local relapse rates than wedge resection. In the special situation of single metastases, our group tended to perform anatomic resections when additional reasons were present, such as larger metastases, an intermediate or central position, or a planned radical lymphadenectomy.

*Lymph node removal*: The extent of lymph node removal during metastasectomy is still a matter of debate. The presence of lymph node metastases is strongly associated with impaired survival [10,26]. Therefore, sampling or dissection of lymph nodes is necessary in order to predict patient prognosis. In our study, we demonstrated that the detection of lymph node involvement was significantly higher when anatomical resection was performed.

To optimize oncological outcomes, it is essential to consider risk factors for local recurrence—such as tumor size, number of metastases, and molecular profile—and to prioritize sufficient safety margins alongside comprehensive lymphadenectomy. Anatomic resection, when feasible, should be strongly considered, especially in patients with identified risk factors, to reduce the likelihood of local recurrence and improve long-term survival.

To understand why we hypothesize that anatomical resection might be the optimal surgical approach for treating single pulmonary colorectal cancer (CRC) metastases, it is essential to consider some fundamental principles. At the time of primary CRC diagnosis, circulating tumor cells (CTCs) are likely present in the bloodstream [27]. Many of these CTCs pass through the pulmonary capillaries, but some become lodged within the lung tissue, where they can initiate the formation of metastases. Consequently, at the time of pulmonary metastasectomy, detectable metastases are present, along with occult micrometastases that may not be visible but can begin to grow shortly after surgery. These invisible micrometastases serve as a source for early development of new pulmonary lesions.

Addressing this challenge involves systemic therapy to target micrometastatic disease, and surgical strategies that preserve pulmonary function—such as parenchyma-sparing resections—are often preferred to facilitate potential repeat interventions. However, it is important to recognize that surgical resection alone cannot prevent the emergence of new metastases, either within the lung or in other organs. The primary goal of pulmonary metastasectomy, therefore, is to remove detectable lesions and prevent local intrapulmonary recurrence at the resection margins.

Furthermore, CRC lung metastases can also release CTCs, which may seed distant metastases outside the lung [28]. It is also well established that CRC lung metastases can metastasize further [9]. Removing the primary source of CTCs through metastasectomy could potentially improve overall survival (OS) by reducing further dissemination. Over time, occult lung metastases may be eliminated or become less likely; the longer the disease-free interval (DFI), the lower the risk of additional micrometastases. This suggests that a single metastasis detected after a prolonged DFI may represent a localized “*early-stage colorectal lung cancer*,” similar to a primary non-small cell lung cancer, rather than widespread metastatic disease.

We assume that a longer persistence of this “primary colorectal lung cancer” increases the risk of local spread, including features such as spread through air spaces (STAS), lymphatic invasion (L1), vascular invasion (V1), and regional lymph node involvement. Based on treatment principles for non-small cell lung cancer (NSCLC), the most effective approach in such cases should be an anatomical resection with systematic lymphadenectomy, aiming for complete (R0) resection and thorough staging.

Several studies report favorable outcomes following resection of solitary CRC lung metastases, with disease-free survival (DFS) and OS rates ranging from 50% to over 70% [5,11,29,30]. These survival rates are comparable to those observed in stage I NSCLC [31,32], raising the question of whether such metastases should be regarded as a distinct clinical entity—similar to an early-stage lung cancer—rather than traditional stage IV disease. This perspective could shift the surgical paradigm from parenchyma-sparing procedures towards more radical, anatomically oriented resections.

Our data indicate that anatomic resection achieves wider pathological margins and is associated with fewer local margin recurrences, whereas overall and recurrence-free survival did not differ between anatomic and non-anatomic approaches. Importantly, tumor recurrence and intrapulmonary tumor recurrence significantly impaired survival outcomes, which was also reported by others [11]. However, we did not find conclusive evidence that anatomical resection directly improves OS or RFS. By definition, following a lobectomy or segmentectomy, a same-site margin recurrence within the removed lobe/segment cannot occur; however, new intrapulmonary lesions outside the resection field, nodal relapse, or distant recurrence remain possible and ultimately dominate prognosis. Furthermore, some patients with local margin recurrence after wedge resection had repeated resections so that their prognosis approximated those without disease recurrence. This helps to explain why a reduction in margin recurrence with anatomical resection did not translate into a survival advantage in our cohort.

In contrast, non-anatomic (wedge/laser) resections prioritize parenchymal preservation and often facilitate repeat metastasectomy if local or new pulmonary recurrences arise. This “salvageability” is clinically relevant in patients with a higher a priori risk of further pulmonary disease (e.g., short DFI, adverse molecular profile, multifocal disease). Taken together, these considerations support a patient-tailored strategy.

A pragmatic approach may be as follows: in centrally located or larger lesions where an adequate margin (e.g., margin-to-tumor ratio ≈ 1) cannot be achieved by wedge, or when nodal involvement is suspected, anatomic resection with systematic nodal dissection should be preferred to optimize local control and staging. Conversely, in patients with higher risk of subsequent intrapulmonary spread—based on clinical course or biology—parenchyma-sparing resection may be prioritized to preserve options for repeated interventions. Future prospective studies should evaluate these trade-offs explicitly, incorporate standardized margin definitions and report nodal management, and include competing-risk analyses to quantify how improvements in local control interact with patterns of failure and survival.

In conclusion, our findings support the notion that radical anatomical resection combined with systematic lymphadenectomy offers a promising approach for achieving durable local disease control in patients with single CRC lung metastases, especially for lesions intermediate or in the periphery of the lung.

### Strengths and Limitations

This study has several limitations. Firstly, it is a retrospective analysis involving a relatively small cohort of patients. Additionally, the study encompasses data from four experienced centers, each employing different surgical concepts for CRC lung metastasectomy. For example, surgeons in Coswig and Loewenstein primarily favored laser resections, whereas those in Hemer and Freiburg preferred stapler resections and anatomical resections, particularly for centrally or intermediately located metastases. Furthermore, missing data on important risk factors—such as minimum safety margins—limits the statistical significance and interpretability of some results. Molecular alterations, which could provide valuable prognostic information, were not available for analysis in this cohort; also, CEA was rarely available and could not be included in the analysis of risk factors. We did not systematically collect data on the possible impact of chemotherapy or immunotherapy, so this remains an unanswered question. Despite these limitations, the variation in management strategies reflects real-world clinical practice and unselected patient populations. This heterogeneity can also be viewed as a strength, as it enhances the generalizability of the findings to routine clinical settings.

## 5. Conclusions

The present multicentre evaluation was conducted with the objective of ascertaining the potential benefit of anatomic resection in comparison with non-anatomic resection for singular colorectal cancer (CRC) lung metastases. Evidence has been presented that indicates that anatomical resection is associated with more complete resections (R0), larger safety margins, a higher number of lymph nodes removed, more frequent radical lymphadenectomies and fewer local intrapulmonary recurrences. However, no substantial prolongation of either the OS or the RFS was observed in the post-anatomical resection group. The recurrence of tumor at any site was observed in 54% of cases, and at intrapulmonary margin in 17.2% with a certain overlap. The potentially beneficial effect on OS or RFS of anatomical instead of non-anatomical resection perished in proportion to the effect of distant recurrence and new lung metastases. However, anatomic resection may be considered in a highly selected group of patients with low risk of distant recurrence, as indicated by metachronous, larger, solitary metastasis with a prolonged disease-free interval and no lymph node involvement. The presence of any tumor recurrence is indicative of a significantly reduced OS and RFS. Consequently, measures aimed at preventing tumor recurrence should be advantageous, and in certain cases, it may be the augmentation of local radicality.

## Figures and Tables

**Figure 1 cancers-17-03299-f001:**
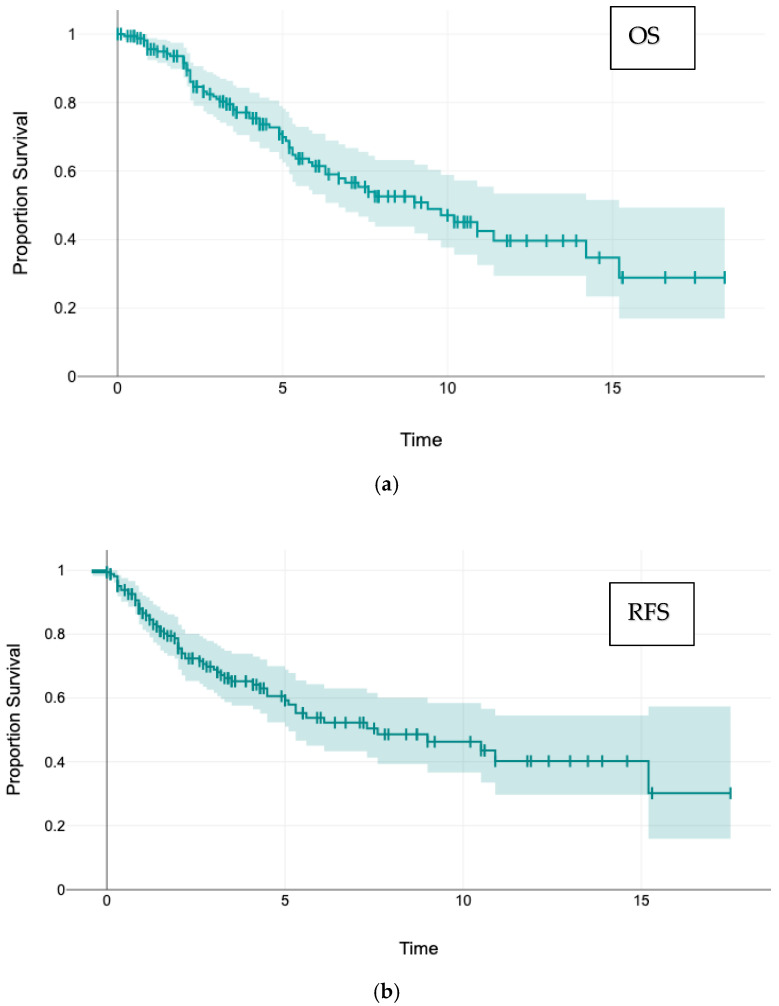
Postmetastasectomy overall survival (OS) and recurrence-free survival (RFS) Kaplan–Meier curves including overlap of 95% CI: (**a**) Overall Survival: 5-year OS 70%, 10-year 47%, median 9.0 years; (**b**) Recurrence-free survival: 5-year RFS 59%, 10-year 46%, median 7.3 years.

**Figure 2 cancers-17-03299-f002:**
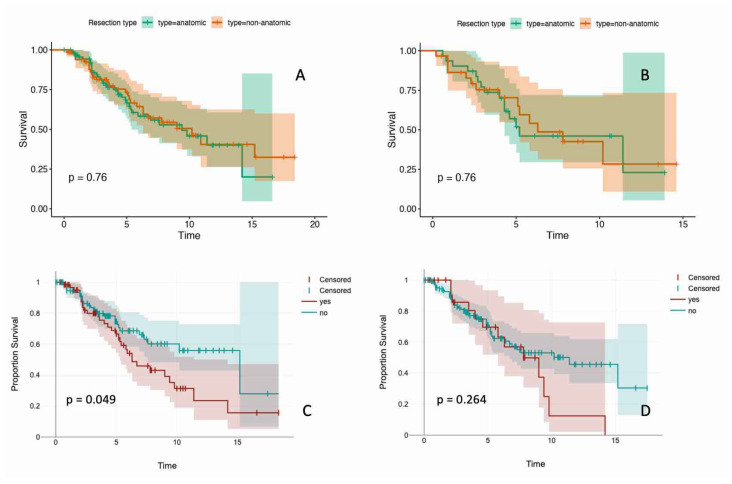
Survival and recurrence after pulmonary metastasectomy. Survival curves and log-rank tests. (**A**) Postmetastasectomy OS of the whole cohort divided for anatomic and non-anatomic resection. (**B**) Postmetastasectomy OS of the intermediate and peripheral metastasis group after PSM divided for anatomic and non-anatomic resection. (**C**) OS with and without intrapulmonary metastasis recurrence. (**D**) OS with and without local intrapulmonary margin recurrence.

**Figure 3 cancers-17-03299-f003:**
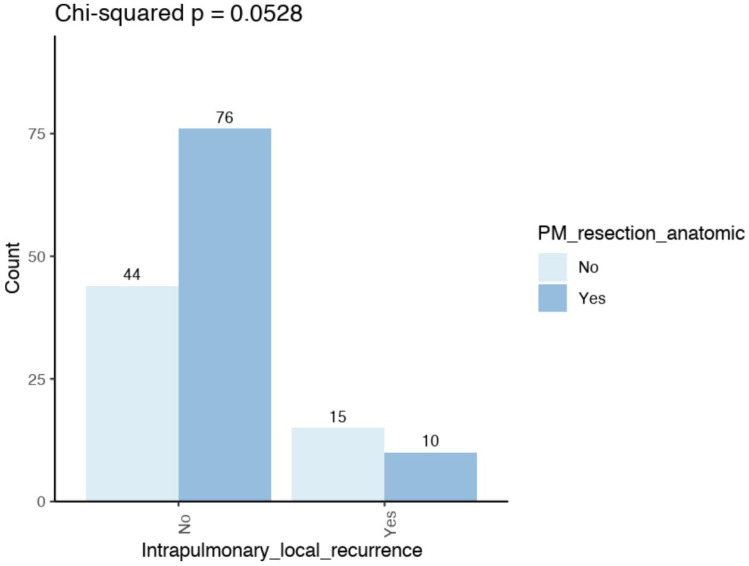
Intrapulmonary local margin recurrence after anatomic or non-anatomic resection concerning the whole cohort.

**Table 1 cancers-17-03299-t001:** Patients demographics.

Variable		*N* (Valid Values)	(% Valid Values)
Gender	Male	104	62.6
	Female	62	37.7
Age (y)		166	100
		median 69, mean 67 (39.2–92.6)	
Primary tumor	Rectum	98	59
	Colon	68	41
Lymph node involvement primary tumor	No	80	53
	Yes	70	47
Resected liver metastasis	No	129	77.7
	Yes	37	22.3
First pulmonary metastasis	Metachronous	144	86.7
	Synchronous	22	13.3
Operative access	Open thoracotomy	126	75.9
	VATS	40	24.1
Extent of resection	Enucleation	41	24.7
	Atypical Resection	32	19.3
	Segmentectomy	36	21.7
	>1 Segments	8	4.8
	Lobectomy	42	25.3
	Bilobectomy	7	4.2
Amount of lung tissue removed	Anatomic Resection	93	56.0
	Non-anatomic Res.	73	44.0
Resection method	Staplers	115	71.0
	Laser resection	45	27.8
	Cautery	2	1.2
Number of specimens removed	1 specimen	123	74.1
	2 specimens	25	15.1
	3 specimens	15	9.0
	>3 specimens	3	1,8
Maximum size of the metastasis (mm)		158	95.2
		median 21.0; mean 27.3 (4–115)	
Minimum safety margin (mm)		106	63.9
		median 5.0; mean 9.9 (0–50)	
Disease-free interval (years)		166	100
		median 2.7; mean 3.2 (0–12.9)	
Follow-up (years)		166	100
		median 4.3; mean 5.3 (0–18.4)	
Postoperative complications (Clavien–Dindo)		165	99.4
	No complication	80	48.5
	Grade I	56	33.9
	Grade II	22	13.3
	Grade IIIa	4	2.4
	Grade IIIb	3	1.8
Laterality	Left	85	51.2
	Right	81	48.8
Affected lobe	Lower lobe	86	51.8
	Upper lobe	62	37.4
	Middle lobe	18	10.8
Resection margin	R0	159	95.8
	R1	7	4.2
Lymphadenectomy	None	29	17.6
	Sampling	79	47.9
	Radical	57	34.5
Number of removed lymph nodes		162	97.6
		median 4; mean 8.2 (0–42)	
Lymph node involvement	N0	147	90.7
	N1	9	5.6
	N2	6	3.7
Position of the metastasis	Peripherical	77	47.0
	Intermediate	35	21.3
	Central	52	31.7
Last status	Alive	102	61.4
	Deceased	64	38.6
Any tumor recurrence	Yes	87	54.0
	No	74	46.0
Tumor recurrence within the lung	Yes	62	38.5
	No	99	61.
Intrapulmonary local margin recurrence	Yes	25	17.2
	No	120	82.8
Intrathoracic lymph node recurrence	Yes	14	10.1
	No	124	89.9
Location of distant recurrence	None	81	48.8
	Unknown	44	26.5
	Primary CRC	13	7.8
	Brain	9	5.4
	Liver	7	4.2
	Multiple sites	7	4.2
	Bone	3	1.8
	Others	2	1.2

**Table 2 cancers-17-03299-t002:** Risk factors for local intrapulmonary margin recurrence, whole cohort (*n* = 145).

Variable		*n*	Intrapulmonary Local Recurrence (%)	Chi-Squared
Gender	Male	89	19 (21.4)	
	Female	56	6 (10.7)	0.154
Primary	Rectum	86	18 (20.9)	
	Colon	59	7 (13.5)	0.232
Resected liver metastasis	no	110	18 (16.4)	
	yes	35	7 (38.9)	0.811
Appearance of metastases	Metachronous	126	21 (16.7)	
	Synchronous	19	19 (26.7)	0.884
Operative access	Thoracotomy	107	19 (17.8)	
	VATS	38	6 (15.8)	0.979
Mode of resection	Non-anatomic	59	15 (25.4)	
	Anatomic	86	10 (11.6)	0.052
Technique of resection	Stapler resection	106	17 (16.0)	
	Laser resection	33	8 (24.2)	
	Cautery	2	0 (0)	0.450
Laterality	Right side	74	14 (18.9)	
	Left side	71	11 (15.5)	0.744
Affected lobe	Upper lobe	56	9 (16.1)	
	Middle lobe	16	2 (14.3)	
	Lower lobe	73	14 (19.2)	0.780
Resection margin	R0	138	23 (16.7)	
	R1	7	2 (28.6)	0.764
Lymphadenectomy	None	25	6 (24.0)	
	Sampling	67	15 (22.4)	
	Radical	53	4 (7.6)	0.063
Lymph node involvement	N0	129	23 (17.8)	
	N1	8	1 (12.5)	
	N2	6	0 (0)	
	N3	1	1 (100)	0.104
Position of the metastasis	Peripherical	65	11 (16.9)	
	Intermediate	33	4 (12.1)	
	Central	46	10 (21.7)	0.534
Last status	Alive	92	12 (13.0)	
	Deceased	53	13 (24.5)	0.125
Intrathoracic lymph node recurrence	No	124	15 (12.1)	
	Yes	14	7 (50)	0.001
	No local margin recurrence Mean/median (mm)		Local margin recurrence Mean/median (mm)	
Maximum size of the metastasis	27.4/20.0		23.2/21.0	0.212
Minimum safety margin	10.7/5.0		4.5/3.0	<0.001
Number of removed lymph nodes	9.3/6.0		4.5/3.0	<0.001
Age (years)	66.7/68.8		69.5/70.5	0.148
Disease-free interval (years)	3.1/2.5		4.3/3.6	0.101

**Table 3 cancers-17-03299-t003:** Risk factors for intrathoracic lymph node recurrence (*p* < 0.1).

Parameter		*n*	Intrapulmonary Lymph Node Recurrence (%)	Chi-Squared
Primary	Rectum	82	12 (14.6)	
	Colon	56	2 (3.6)	0.068
Lymph node involvement	N0	122	10 (8.2)	
	N1	8	3 (37.5)	
	N2	6	1 (16.7)	
	Nx	1	0 (0)	0.060
Last status	Alive	87	4 (4.6)	
	Deceased	51	10 (19.6)	0.012
Pulmonary tumor recurrence	No	84	2 (2.4)	
	Yes	42	12 (28.6)	<0.001
Intrapulmonary local margin recurrence	No	116	7 (6.0)	
	Yes	22	7 (31.8)	0.001

**Table 4 cancers-17-03299-t004:** Correlations between anatomic and non-anatomic resections (*p* < 0.1).

Parameter		*n* (%) Anatomic res.	*n* (%) Non-Anatomic res.	Chi^2^
Resection margin	R0	93 (100)	66 (90.4)	
	R1	0 (0)	7 (9.6)	0.007
Lymphadenectomy	None	12 (12.9)	17 (23.6)	
	Sampling	27 (29.0)	52 (72.2)	
	radical	54 (58.1)	3 (4.2)	<0.001
Lymph node involvement	N0	78 (84.8)	67 (98.5)	
	N1	9 (9.8)	0 (0)	
	N2	5 (5.4)	1 (1,5)	0.027
Position of the metastasis	Peripherical	28 (30.1)	49 (69.0)	
	Intermediate	26 (28.0)	9 (12.7)	
	Central	39 (41.9)	13 (18.3)	0.012
Intrapulmonary local recurrence	No	76 (88.4)	44 (74.6)	
	yes	10 (11.6)	15 (25.4)	0.052
		Mean/median	Mean/median	
Age (years)		66.9/69.8	67.2/68.0	0.841
Disease-free interval (Years)		3.5/3.1	3.0/2.3	0.215
Maximum size of the metastasis (mm)		32.9/28.0	19.9/18.0	<0.001
Minimum safety distance (mm)		11.8/7.0	5.5/4.5	<0.001
Number of lymph nodes removed		12.5/12.0	2.7/2.0	<0.001

**Table 5 cancers-17-03299-t005:** Hypothesis testing—anatomical vs. non-anatomical resection for peripherical and intermediate metastases.

Parameter		No Local Margin Recurrence	Local Margin Recurrence	95% CI	OR	*p*
Mode of resection	Anatomical	47	4			
	Non-anatomical	42	11	1.01–7.18	2.63	0.042
		**no intrath. LN** **recurrence**	**intrathorac LN** **recurrence**			
Mode of resection	Anatomical	48	3			
	Non-anatomical	46	5	0.47–6.04	1.68	0.392

## Data Availability

The data presented in this study are available on request from the corresponding author due to different safety requests from the contributing institutions.

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
