# Peer review of "Pulmonary Metastasectomy: A Multicenter Comparison of Wedge Resection Versus Anatomic Resection for Single Metastases of Colorectal Cancer"

_cancers, 2025, doi:10.3390/cancers17203299_

Round 1
Reviewer 1 Report
Comments and Suggestions for Authors
This paper highlights an essential point in surgery for pulmonary metastasis and emphasizes the importance of achieving sufficient surgical margins. However, there are several issues that should be addressed before publication.
- The proportion of anatomical resections appears to be relatively high. The indication criteria for performing anatomical resection in cases of solitary pulmonary metastasis should be clearly described. How were patients selected for anatomical resection?
- Since pulmonary metastases frequently develop pulmonary recurrence, resection for metachronous ipsilateral pulmonary metastasis may be technically challenging. In addition, wedge resection can preserve more normal lung parenchyma, whereas anatomical resection may have the drawback of sacrificing greater lung volume. These points should be discussed.
- I think the benefit of lymph node sampling or dissection during pulmonary metastasectomy remains controversial. It is not clear whether anatomical resection provides any survival advantage in terms of overall survival or recurrence-free survival. Further clarification on this point is warranted.
Author Response
Comment 1: The proportion of anatomical resections appears to be relatively high. The indication criteria for performing anatomical resection in cases of solitary pulmonary metastasis should be clearly described. How were patients selected for anatomical resection?
Answer 1: Thank you for pointing this out. Indeed, the number of anatomical resections is high when pursuing the goal of preservation of lung function. The four contributing centers have different approaches with the common goal of keeping enough safety margins. Furthermore, the median size of the lesions was 21mm and if not located subpleurally anatomic resection was preferred to keep sufficient margins. As only one lesion had to be removed, parenchyma preservation was not the first goal.
Changes in the manuscript 1 (line 109-115): The surgery was performed in accordance with local standards and the assessment of the responsible surgeon. As only one lesion had to be removed, sufficient safety margins were prioritised over tissue preservation. When the laser was used, the surgeon stated that complete resection had been achieved. If R1 was stated by the pathologist, it was included in the database even after laser resection. An anatomic resection was performed when the lesion was larger than 20 mm or centrally located.
See also (line 533-535): Operative procedures were performed according to local standards, with no standardized protocol across centers. This resulted in variability in techniques, including the use of laser and minimally invasive approaches.
And (line 591-593): The choice of resection method was primarily dictated by intraoperative assessment and surgeon preference, which resulted in a moderate local recurrence rate of 17.2% in our study.
Comment 2: Since pulmonary metastases frequently develop pulmonary recurrence, resection for metachronous ipsilateral pulmonary metastasis may be technically challenging. In addition, wedge resection can preserve more normal lung parenchyma, whereas anatomical resection may have the drawback of sacrificing greater lung volume. These points should be discussed.
Answer 2: We fully agree with this statement. But also, patients with single metachronous CRC metastases have the best prerequisite to be cured. In our series, 46% of patients did not face any tumor recurrence. For them it is of major interest to prevent local intrapulmonary recurrence and (only) for them, we think anatomic resection could be beneficial.
Changes in the manuscript 2 (line 603-611): In our series, the number of anatomical resections was higher than that of non-anatomical resections (93 versus 73). Wedge resection is widely accepted as the standard method for pulmonary metastasectomy, mainly because the amount of resection is not related to survival in most cases. Wedge resection is also associated with lower morbidity, and lung-sparing resection allows for eventual redo surgery. Conversely, segmentectomy and even lobectomy are associated with lower local relapse rates than wedge resection. In the special situation of single metastases, our group tended to perform anatomic resections when additional reasons were present, such as larger metastases, an intermediate or central position, or a planned radical lymphadenectomy.
See also line 676-680.
Comment 3: I think the benefit of lymph node sampling or dissection during pulmonary metastasectomy remains controversial. It is not clear whether anatomical resection provides any survival advantage in terms of overall survival or recurrence-free survival. Further clarification on this point is warranted.
Answer 3: Yes, this is an important point. To give an answer to this question, we initiated this study. We could demonstrate in former publications, that CRC pulmonary metastases tend to spread in the lung in a similar manner as primary adenocarcinomas of the lung: STAS, L1, V1, Spread into lymph nodes (doi: 10.1093/ejcts/ezx063). These patterns of local spread might be the source for later local recurrence. See line 635-638. Large safety margins or even anatomic resections with lymph node removal would be the adequate consequence. Furthermore, as presented in other studies, lymph-node metastasis is a significant prognostic factor predicting poor outcome. Lymph-node sampling or dissection is therefore warranted to predict patient prognosis.
Changes in the manuscript 3 (line 612-616): Lymph node removal: The extent of lymph node removal during metastasectomy is still a matter of debate. The presence of lymph node metastases is strongly associated with impaired survival [10, 26]. Therefore, sampling or dissection of lymph nodes is necessary in order to predict patient prognosis. In our study, we demonstrated that the detection of lymph node involvement was significantly higher when anatomical resection was performed.
And the following reference was added:
- van Dorp M, Bousema JE, Torensma B, Dickhoff C, van den Broek FJC, Schreurs WH, Gonzalez M, Kazemier G, Heineman DJ. Pulmonary metastasectomy with lymphadenectomy for colorectal pulmonary metastases: A systematic review. Eur J Surg Oncol. 2022 Jan;48(1):253-260. doi: 10.1016/j.ejso.2021.09.020.
Reviewer 2 Report
Comments and Suggestions for Authors
Dear Editor and Authors,
As a practicing thoracic surgeon this reviewer had the pleasure to review this work titled "Pulmonary metastasectomy: a multicenter comparison of wedge resection versus anatomic resection for single metastases of colorectal cancer" by Dr. Stefan Welter and his colleagues.
In this work the authors present their experience from five centers on single nodule pulmonary metastasectomy for colorectal cancer. They perform an analysis of 166 patients comparing limited resections, wedge (but also enucleations) with anatomical segmentectomy or lobectomy! They utilize modern analytics including propensity score matching of the groups!! Consequently they demonstrated that anatomical resection produce better outcomes in terms of overall mortality and disease free survival as well as local recurrence!!
The study is quite interesting, it has a modereate sample size but it is multi-institutional and despite its retrospective nature the methodology and analysis utilized is robust. The manuscript is well written and presented with clear and undestandable language which helps readers comprehend the work adequately.
I do have some points which I would like to be addressed before this reviewer could recommend the publication of this work. Specifically;
Comments:
- The introduction is slightly large including the presentation of a number of studies from the literature and their outcomes! I would suggest keeping the main ones to establish the premise of the study and moving the rest in the discussion section!
- Where the data gathered and analyzed from all the centers' databases compatible and utilizing similar parameters/definitions? Did all centers collect similar variables and data? Please address this in the methods section.
- What where the inclusion and exclusion criteria for patient data extraction and inclusion in the master database? Did each Center collate and send it their own data based on pre-determined parameters or where all patients from each center included in a large pool and then extraction was performed. The reason this is needed to be presented is to prevent the notion of bias from individual centers/researchers (i.e. each center pulled their best cases and submitted those as opposed to a large master pool!!).
- Was a power / sample size calculation performed to assess if the number of patients is adequate to produce meaningful results?
- How was follow up performed and who performed it? Surgeons or oncologists? Was follow up adequate?
- Why does the sample contain such a large number of open metastasectomies for single nodule disease as opposed to the less invasive VATS? Is there some operative policy for open metastasectomy (some centers advocate it) in individual centers or was it surgeon preference?? What is the distribution of open vs. VATS amongst centers?
- Why was enucleation performed for malicnancy?? Also why where multiple segments removed if it was a single nodule disease? Why where lobectomies and bilobectomies performed for metastatic disease??
- What was the average and median size of the metastatic tumors??
- If this was metastasectomy for single nodule why do the authors report resection of 2, 3 up to 7 specimens??
- What is vital status - maybe use another term!!
- Why report distant metastatic sites since the analysis supposedly focuses on local reccurence??
- In truth the aims of the study don't correspond with the reported data as multiple techniques are pooled together and compared each with their own morbidity/mortality and outcomes!!
- How was propensity score matching performed? What where the matching parameters used?
- Why isn't data on oncological therapies (classical chemo, immunotherapy, radiation therapy) presented and taken into account in regards to survival/recurrence data?
- Patients who received redo metastasectomies should have been exclused from the analysis because of bias and survival outcome prolongation with the second (?third?) procedure!!
In conclusion, althouth this study aims to address quite an interesting subject it fails to do so in a clear and unbiased maner! I am reluctunt to recommend publication in its present form!! Thank you.
Author Response
Comment 1: The introduction is slightly large including the presentation of a number of studies from the literature and their outcomes! I would suggest keeping the main ones to establish the premise of the study and moving the rest in the discussion section!
Answer 1: Dear reviewer. We are very thankful for your comprehensive valuation of our paper and appreciate every comment. We understand that the introduction is rather long, but we feel it very important to give the reader necessary scientific background to be prepared for the following detailed results.
Comment 2: Where the data gathered and analyzed from all the centers' databases compatible and utilizing similar parameters/definitions? Did all centers collect similar variables and data? Please address this in the methods section.
Answer 2: This is an important point. We constructed a common entry mask and the anonymized values were included from each center. Please see text line 120-130
Comment 3: What where the inclusion and exclusion criteria for patient data extraction and inclusion in the master database? Did each Center collate and send it their own data based on pre-determined parameters or where all patients from each center included in a large pool and then extraction was performed. The reason this is needed to be presented is to prevent the notion of bias from individual centers/researchers (i.e. each center pulled their best cases and submitted those as opposed to a large master pool!!).
Answer 3: Thank you for this very precise question to the methodology. We created a common entry mask with predefined parameters. Every center received the empty mask and confirmed, to entry every case with single CRC lung metastasis. This resulted in 181 cases consolidated into one database by the corresponding author. Than, 15 cases had to be excluded for incomplete data. All coauthors confirmed the delivery of all single CRC lung metastases.
Changes in the text (line 120-124): Each center received a standardized entry form containing all the relevant parameters and submitted every case involving resection of single colorectal cancer (CRC) metastases after anonymization. All anonymized local data were then consolidated into one entry mask containing the following parameters:
Comment 4: Was a power / sample size calculation performed to assess if the number of patients is adequate to produce meaningful results?
Answer 4: No, a sample size calculation was not performed as this retrospective study was planned for hypothesis generation. We plan to discuss a lager prospective database within the german society of thoracic surgeons in the future.
Comment 5: How was follow up performed and who performed it? Surgeons or oncologists? Was follow up adequate?
Answer 5: Follow-up is usually performed in Colo-rectal Cancer centers, but patients are requested to provide their CT-scans of the chest to the thoracic surgeon. See line 139-142: All follow-up CT scans with a suspicion for metastasis recurrence in the lung were reviewed by an experienced surgeon from each contributing site. Written CT reports from radiologists stating “no tumor recurrence” in the chest were accepted. Unfortunately, due to the retrospective nature of this analysis, a systematic examination of every patient by a thoracic surgeon is not possible.
Comment 6: Why does the sample contain such a large number of open metastasectomies for single nodule disease as opposed to the less invasive VATS? Is there some operative policy for open metastasectomy (some centers advocate it) in individual centers or was it surgeon preference?? What is the distribution of open vs. VATS amongst centers?
Answer 6: The philosophy in 2 centers was to perform metastasectomy in an open procedure to always palpate the whole lung. Therefore, two centers (Cos / Löw) had only 3 VATS procedures whereas Hem/Frei performed all the other VATS procedures. The operative access may have an influence on postoperative complications, but we do not think it influences local recurrence. Therefore we would not make any changes in the text.
Comment 7: Why was enucleation performed for malicnancy?? Also why where multiple segments removed if it was a single nodule disease? Why where lobectomies and bilobectomies performed for metastatic disease??
Answer 7: In Germany, laser enucleation for lung metastases is a very popular method. The first laser enucleations were performed in one centre (Coswig) in the 1990s. 'Multiple segments' is misleading. It means two or three neighbouring segments for one metastasis. Lobectomies or bilobectomies were necessary for central lesions. Thirty-six metastases were 40 mm or larger, justifying larger resections.
Changes in the text 7 (line 200): >1 segment
Comment 8: What was the average and median size of the metastatic tumors??
Answer 8: see line 216: Maximum size of the metastasis (mm): median 21.0; mean 27.3 (4 – 115)
Comment 9: If this was metastasectomy for single nodule why do the authors report resection of 2, 3 up to 7 specimens??
Answer 9: The intention of every intervention was to resect every suspicious lesion. Therefore sometimes more than on nodule was resected but specified as non-malignant by the pathologist. Finally all patients had only one confirmed metastasis
Comment 10: What is vital status - maybe use another term!!
Answer 10: Thank you for this hint. We will use “last status”. Corresponding changes are performed.
Comment 11: Why report distant metastatic sites since the analysis supposedly focuses on local reccurence??
Answer 11: Thank you for this notion. While we demonstrate that anatomical resection reduces local recurrence, it cannot improve RFS and OS because the benefits of local recurrence prevention are outweighed by distant recurrence in the lung and other organs. Therefore, while the location of distant recurrence may be less important, the occurrence of distant metastases as an important prognostic factor is of great importance in our view.
Comment 12: In truth the aims of the study don't correspond with the reported data as multiple techniques are pooled together and compared each with their own morbidity/mortality and outcomes!!
Answer 12: We agree, that different techniques arise some questions. But as operative mortality was 0% and Clavien-Dindo complication rate is low. We do not think, that different access and different resection methods affect the patients too much. The main outcome: metastasis recurrence at the resection margin was compared between anatomic and non-anatomic resection. And the four different centers represent the whole spectrum of real life metastasis management. We already mentioned this bias in the limitations section line 688-692)
Comment 13: How was propensity score matching performed? What where the matching parameters used?
Answer 13: PS matching was performed for 112 cases with intermediate and peripheral metastases using the significant differences (p < 0.1) between the anatomic and non-anatomic group as matching parameters.
Comment 14: Why isn't data on oncological therapies (classical chemo, immunotherapy, radiation therapy) presented and taken into account in regards to survival/recurrence data?
Answer 14: We agree, that this might also be an important factor, but we discussed this before the study protocol was finalized. We found it too difficult to categorize “adjuvant”, additional chemotherapy or palliative chemo in case of another tumor recurrence. Finally almost every patient had chemo at some time in their treatment journey. But this seems to be an important question for further projects.
Changes in the text 14 (697-698): We did not systematically collect data on the possible impact of chemotherapy or immunotherapy, so this remains an unanswered question.
Comment 15: Patients who received redo metastasectomies should have been exclused from the analysis because of bias and survival outcome prolongation with the second (?third?) procedure!!
Answer 15: We fully agree that redo surgery prolongs survival and thus minimises the benefit of anatomical resection during the initial intervention. However, these cases are counted as local margin recurrences, and the date of the first recurrence is included in the RFS. Therefore, the main question of the study is not biased; only the OS difference between anatomical and non-anatomical resection is blurred. Please see line 663: Furthermore, some patients with local margin recurrence after wedge resection had repeated resections so that their prognosis approximate those without disease recurrence. This helps to explain why a reduction in margin recurrence with anatomical resection did not translate into a survival advantage in our cohort.
Round 2
Reviewer 2 Report
Comments and Suggestions for Authors
Dear Editor and Authors,
Thank you for asking me to re-evaluate this work. I read it with interest and also noted the response the authors gave to the comments. Although, they have very thoroughly and logically addressed the issues they did not implement or mention or explain (like they did in the response to the reviewer) their comments/logic/though process/reason ect in their text. In reality they only responsed to the comments BUT did not make any real changes/revision in their text. I therefore recommend that they utilize their commentary to the reviewers and add it into their manuscript to strengthen it!!
Thank you.
Author Response
Round 2: Reviewers Comments
Reviewer 2:
Comment 1, round 2: Thank you for asking me to re-evaluate this work. I read it with interest and also noted the response the authors gave to the comments. Although, they have very thoroughly and logically addressed the issues they did not implement or mention or explain (like they did in the response to the reviewer) their comments/logic/though process/reason ect in their text. In reality they only responsed to the comments BUT did not make any real changes/revision in their text. I therefore recommend that they utilize their commentary to the reviewers and add it into their manuscript to strengthen it!!
Answer 1 round 2: Dear reviewer, thank you very much for the repeated evaluation and review of our manuscript. We have made a lot of additions, but there might be some more changes necessary to address all comments. The following answers are from the first round with additional comments to the second round.
Comment 1: The introduction is slightly large including the presentation of a number of studies from the literature and their outcomes! I would suggest keeping the main ones to establish the premise of the study and moving the rest in the discussion section!
Answer 1: Dear reviewer. We are very thankful for your comprehensive valuation of our paper and appreciate every comment. We understand that the introduction is rather long, but we feel it very important to give the reader necessary scientific background to be prepared for the following detailed results.
Answer round 2: we moved a long passage from the introduction into the discussion section (red writing)
Changes in the text: line 322-337
Comment 2: Where the data gathered and analyzed from all the centers' databases compatible and utilizing similar parameters/definitions? Did all centers collect similar variables and data? Please address this in the methods section.
Answer 2: This is an important point. We constructed a common entry mask and the anonymized values were included from each center. Please see text line 120-130
Answer round 2: We added text in line 110-114, but forgot to make it red.
Changes in the text: line 110-114: "Each centre received a standardised entry form containing all the relevant parameters and submitted every case involving resection of single colorectal cancer (CRC) metastases after anonymisation. All anonymised local data were then consolidated into one entry mask containing the following parameters:"
Comment 3: What where the inclusion and exclusion criteria for patient data extraction and inclusion in the master database? Did each Center collate and send it their own data based on pre-determined parameters or where all patients from each center included in a large pool and then extraction was performed. The reason this is needed to be presented is to prevent the notion of bias from individual centers/researchers (i.e. each center pulled their best cases and submitted those as opposed to a large master pool!!).
Answer 3: Thank you for this very precise question to the methodology. We created a common entry mask with predefined parameters. Every center received the empty mask and confirmed, to entry every case with single CRC lung metastasis. This resulted in 181 cases consolidated into one database by the corresponding author. Than, 15 cases had to be excluded for incomplete data. All coauthors confirmed the delivery of all single CRC lung metastases.
Changes in the text (line 110-114): Each center received a standardized entry form containing all the relevant parameters and submitted every case involving resection of single colorectal cancer (CRC) metastases after anonymization. All anonymized local data were then consolidated into one entry mask containing the following parameters:
Answer 3 round 2: see changes in the text line 110-114 (Comment 2)
Comment 4: Was a power / sample size calculation performed to assess if the number of patients is adequate to produce meaningful results?
Answer 4: No, a sample size calculation was not performed as this retrospective study was planned for hypothesis generation. We plan to discuss a lager prospective database within the german society of thoracic surgeons in the future.
Answer 4 round 2: we added one sentence in the method section
Changes in the text (Line 139-140): “As the study was retrospective, no sample size calculation was performed in advance.”
Comment 5: How was follow up performed and who performed it? Surgeons or oncologists? Was follow up adequate?
Answer 5: Follow-up is usually performed in Colo-rectal Cancer centers, but patients are requested to provide their CT-scans of the chest to the thoracic surgeon. See line 139-142: All follow-up CT scans with a suspicion for metastasis recurrence in the lung were reviewed by an experienced surgeon from each contributing site. Written CT reports from radiologists stating “no tumor recurrence” in the chest were accepted. Unfortunately, due to the retrospective nature of this analysis, a systematic examination of every patient by a thoracic surgeon is not possible.
Answer 5 round 2: Additions in the methods section were added.
Changes in the text (line 130-134): “Follow-up is usually performed in colorectal cancer centers, but patients were requested to provide their chest CT scans to the thoracic surgery department. All follow-up CT scans showing a possible recurrence of metastasis in the lung were reviewed by an experienced surgeon from each contributing site. Written CT reports from radiologists stating 'no tumor recurrence' in the chest were accepted. Unfortunately, due to the retrospective nature of this analysis, it is not possible to examine every patient systematically. ”
Comment 6: Why does the sample contain such a large number of open metastasectomies for single nodule disease as opposed to the less invasive VATS? Is there some operative policy for open metastasectomy (some centers advocate it) in individual centers or was it surgeon preference?? What is the distribution of open vs. VATS amongst centers?
Answer 6: The philosophy in 2 centers was to perform metastasectomy in an open procedure to always palpate the whole lung. Therefore, two centers (Cos / Löw) had only 3 VATS procedures whereas Hem/Frei performed all the other VATS procedures. The operative access may have an influence on postoperative complications, but we do not think it influences local recurrence. Therefore we would not make any changes in the text.
Answer 6 round 2: as there is no international recommendation to perform metastasectomy in a VATS procedure, we like to make no changes in the text.
Comment 7: Why was enucleation performed for malicnancy?? Also why where multiple segments removed if it was a single nodule disease? Why where lobectomies and bilobectomies performed for metastatic disease??
Answer 7: In Germany, laser enucleation for lung metastases is a very popular method. The first laser enucleations were performed in one centre (Coswig) in the 1990s. 'Multiple segments' is misleading. It means two or three neighbouring segments for one metastasis. Lobectomies or bilobectomies were necessary for central lesions. Thirty-six metastases were 40 mm or larger, justifying larger resections.
Changes in the text 7 (line 200): >1 segment
Answer 7 round 2: One sentence was added in the discussion section:
Changes in the text 7 (line 319-321): "As 36 metastases were larger than 4cm, and many metastases had a central position, the relatively high number of anatomical resections in our series may be explained."
Comment 8: What was the average and median size of the metastatic tumors??
Answer 8: see line 216: Maximum size of the metastasis (mm): median 21.0; mean 27.3 (4 – 115)
Comment 9: If this was metastasectomy for single nodule why do the authors report resection of 2, 3 up to 7 specimens??
Answer 9: The intention of every intervention was to resect every suspicious lesion. Therefore sometimes more than on nodule was resected but specified as non-malignant by the pathologist. Finally all patients had only one confirmed metastasis
Answer 9 round 2: minor changes in the text were made.
Changes in the text 9 (line 316-319): "Furthermore, every suspicious lesion was removed. In some cases, therefore, more than one specimen reached the pathologist. Ultimately, one metastasis was proven in each patient, with no evidence of any other malignancy."
Comment 10: What is vital status - maybe use another term!!
Answer 10: Thank you for this hint. We will use “last status”. Corresponding changes are performed.
Comment 11: Why report distant metastatic sites since the analysis supposedly focuses on local reccurence??
Answer 11: Thank you for this notion. While we demonstrate that anatomical resection reduces local recurrence, it cannot improve RFS and OS because the benefits of local recurrence prevention are outweighed by distant recurrence in the lung and other organs. Therefore, while the location of distant recurrence may be less important, the occurrence of distant metastases as an important prognostic factor is of great importance in our view.
Answer 11, round 2: no further comments
Comment 12: In truth the aims of the study don't correspond with the reported data as multiple techniques are pooled together and compared each with their own morbidity/mortality and outcomes!!
Answer 12: We agree, that different techniques arise some questions. But as operative mortality was 0% and Clavien-Dindo complication rate is low. We do not think, that different access and different resection methods affect the patients too much. The main outcome: metastasis recurrence at the resection margin was compared between anatomic and non-anatomic resection. And the four different centers represent the whole spectrum of real life metastasis management. We already mentioned this bias in the limitations section line 688-692)
Answer 12, round 2 (line 490-494): The following text is already in the limitations section: “Additionally, the study encompasses data from four experienced centers, each employing different surgical concepts for CRC lung metastasectomy. For example, surgeons in Coswig and Loewenstein primarily favored laser resections, whereas those in Hemer and Freiburg preferred stapler resections and anatomical resections, particularly for centrally or intermediately located metastases.“
Comment 13: How was propensity score matching performed? What where the matching parameters used?
Answer 13: PS matching was performed for 112 cases with intermediate and peripheral metastases using the significant differences (p < 0.1) between the anatomic and non-anatomic group as matching parameters.
Answer 13, round 2 (line 163-165): the answer was added in the methods section.
Comment 14: Why isn't data on oncological therapies (classical chemo, immunotherapy, radiation therapy) presented and taken into account in regards to survival/recurrence data?
Answer 14: We agree, that this might also be an important factor, but we discussed this before the study protocol was finalized. We found it too difficult to categorize “adjuvant”, additional chemotherapy or palliative chemo in case of another tumor recurrence. Finally almost every patient had chemo at some time in their treatment journey. But this seems to be an important question for further projects.
Changes in the text 14 (499-500): "We did not systematically collect data on the possible impact of chemotherapy or immunotherapy, so this remains an unanswered question."
Comment 15: Patients who received redo metastasectomies should have been exclused from the analysis because of bias and survival outcome prolongation with the second (?third?) procedure!!
Answer 15: We fully agree that redo surgery prolongs survival and thus minimises the benefit of anatomical resection during the initial intervention. However, these cases are counted as local margin recurrences, and the date of the first recurrence is included in the RFS. Therefore, the main question of the study is not biased; only the OS difference between anatomical and non-anatomical resection is blurred. Please see line 663: Furthermore, some patients with local margin recurrence after wedge resection had repeated resections so that their prognosis approximate those without disease recurrence. This helps to explain why a reduction in margin recurrence with anatomical resection did not translate into a survival advantage in our cohort.
Answer 15, round 2: we do not want to make further changes.